# Defatted Seeds of *Oenothera biennis* as a Potential Functional Food Ingredient for Diabetes

**DOI:** 10.3390/foods10030538

**Published:** 2021-03-05

**Authors:** Zhiqiang Wang, Zhaoyang Wu, Guanglei Zuo, Soon Sung Lim, Hongyuan Yan

**Affiliations:** 1Key Laboratory of Public Health Safety of Hebei Province, College of Public Health, Hebei University, Baoding 071002, China; wzy19970202@163.com; 2Key Laboratory of Medicinal Chemistry and Molecular Diagnosis of Ministry of Education, Institute of Life Science and Green Development, Hebei University, Baoding 071002, China; 3Department of Food Science and Nutrition, Hallym University, 1 Hallymdeahak-gil, Chuncheon 24252, Korea; B16504@hallym.ac.kr (G.Z.); limss@hallym.ac.kr (S.S.L.)

**Keywords:** defatted seeds of *Oenothera biennis*, α-glucosidase, aldose reductase, antioxidant, polyphenols, nutrients

## Abstract

The defatted seeds of *Oenothera biennis* (DSOB) are a by-product of evening primrose oil production that are currently not effectively used. In this study, α-glucosidase inhibition, aldose reductase inhibition, antioxidant capacity, polyphenol composition, and nutritional value (carbohydrates, proteins, minerals, fat, organic acid, and tocopherols) of DSOB were evaluated using the seeds of *Oenothera biennis* (SOB) as a reference. DSOB was an excellent inhibitor of α-glucosidase (IC_50_ = 3.31 μg/mL) and aldose reductase (IC_50_ = 2.56 μg/mL). DSOB also showed considerable antioxidant capacities (scavenging of 2,2-diphenyl-1-picrylhydrazyl, 2,2’-azino-bis(3-ethylbenzothiazoline-6-sulfonic acid, nitric oxide, peroxynitrite, and hydroxyl radicals). DSOB was a reservoir of polyphenols, and 25 compounds in DSOB were temporarily identified by liquid chromatography coupled with electrospray ionization–quadrupole time of flight–mass spectrometry analysis. Moreover, the carbohydrate, protein, and mineral content of DSOB were increased compared to that of SOB. DSOB contained large amounts of fiber and low levels of sugars, and was rich in calcium and iron. These results imply that DSOB may be a potential functional food ingredient for diabetes, providing excellent economic and environmental benefits.

## 1. Introduction

Diabetes mellitus, characterized by abnormal hyperglycemia, affects 463 million patients globally [1], of which 60% have one or more complications, such as diabetic retinopathy, nephropathy, and neuropathy [2]. The results of epidemiological and clinical research show that effective blood glucose control postpones the occurrence of diabetic complications; however, its development in patients with diabetes is certain to happen [3,4]. Furthermore, changes in diet or medication of patients with diabetes often results in nutritional deficiencies, dramatically damaging the health and lowering quality of life [5]. Thus, the health management of patients with diabetes should be concerned with the control of hyperglycemia, prevention of complications, and supplementation of nutrition. α-Glucosidases are a group of intestinal enzymes involved in carbohydrate digestion and are critical targets for the amelioration of postprandial hyperglycemia [6]. Aldose reductase is a pivotal target for preventing the onset and progression of diabetic complications [7]. Moreover, oxidative stress is strongly related to the onset of diabetes and exacerbation of complications [8]. Currently, the increasingly mindful attitudes of consumers to their diet and health has led to the development of new trends, including the widespread use of functional foods. Hence, there is growing interest in developing functional food ingredients that possess effective inhibitory activities against α-glucosidase, aldose reductase, and free radicals, particularly those with nutritional supplementation that may be useful for promoting the health of patients with diabetes.

*Oenothera biennis* (OB), commonly known as evening primrose, is an herbaceous plant of the family *Onagraceae* [9]. In recent decades, oil extracted from the seeds of OB (SOB) has drawn attention because of its high polyunsaturated fatty acid content (γ-linolenic acid) and excellent bioactive properties [10]. Thus, OB is globally cultivated as an industrial oil crop for the production of evening primrose oil (EPO) used in the development of pharmaceuticals, cosmetics, food products, bakery products, and confectionery [11]. The annual global production of SOB has increased 20-fold in the last 20 years, producing kilotons of seeds annually, and is expected to experience robust growth in the foreseeable future [12]. However, 50–55% of SOB generated after EPO manufacture are residues (defatted SOB (DSOB)), resulting in a large waste of resources. The disposal of DSOB seed residues generated by oil extraction is an important issue for related industries, and its use may lead to great economic and environmental benefits, if utilized properly.

SOB is a rich source of not only a valuable oil, containing essential fatty acids, but also of polyphenols, which have shown remarkable bioactivity [13]. Moreover, SOB is also rich in nutrients containing proteins, carbohydrates, minerals, and vitamins, in addition to the oil [14]. However, to date, few studies have evaluated the nutritional value and polyphenol composition of DSOB, nor the biological properties related to diabetes, which limits the potential reuse of DSOB. In this study, to evaluate the potential of DSOB as a functional food ingredient in diabetes, we investigated the inhibitory effects of DSOB on α-glucosidase and aldose reductase for the first time, and evaluated the antioxidant capacities of DSOB by electron transfer assays (1,1-diphenyl-2-picrylhydrazyl (DPPH) and 2,2-azinobis(3-ethylbenzothiazoline-6-sulfonic acid) diammonium salt (ABTS)) and reactive oxygen species (ROS) scavenging assays (hydroxyl radical (HO•), nitric oxide (NO), and peroxynitrite (ONOO^−^)). Moreover, the polyphenol composition and nutritional value of DSOB were analyzed compared to SOB as a reference.

## 2. Materials and Methods

### 2.1. Chemicals

α-Glucosidase from *Saccharomyces cerevisiae*, ascorbic acid, acarbose, ammonium sulfate, *p*-nitrophenyl-α-D-glucopyranoside, Trolox, β-nicotinamide adenine dinucleotide 2-phosphate reduced tetrasodium salt hydrate (NADPH), DL-glyceraldehyde, epalrestat, quercetin, oxalic acid, citric acid, glucose, fructose, galactose, lactose, maltose, sucrose, tocopherols (α-, β-, and γ-), hydrogen peroxide (H_2_O_2_), ABTS, DPPH, Folin and Ciocalteu’s phenol reagent, Griess reagent, and 2,6-di-tert-butyl-4-methylphenol (BHT) were purchased from Sigma-Aldrich (St. Louis, MO, USA). The standards used for the determination of mineral contents, including sodium (Na), potassium (K), calcium (Ca), magnesium (Mg), iron (Fe), manganese (Mn), copper (Cu), and zinc (Zn), were obtained from the Central Iron and Steel Research Institute (Beijing, China). ONOO^−^ was obtained from Cayman Chemical (Ann Arbor, MI, USA). All organic solvents were purchased from Concord Technology (Tianjin, China). Ultrapure water (18.2 MΩ cm) was prepared using a water purification system (Milli-Q; Millipore, Billerica, MA, USA). Unless otherwise stated, all other reagents were purchased from Sigma-Aldrich.

### 2.2. Plant Samples

SOB was purchased from Yiyuan Chinese Herb Medicine Co. Ltd. (Anguo, Hebei, China) in August 2018. Dry ground SOB (1000.21 g) was defatted twice with *n*-hexane over a period of 48 h to obtain DSOB (788.69 g). Voucher DSOB (WLL-2018-01) and SOB (WLL-2018-02) were deposited at the College of Public Health, Hebei University.

### 2.3. Macronutrients, Energy, and Sugars

Macronutrients (carbohydrates, proteins, fats, and ash), energy, and sugars (glucose, fructose, galactose, lactose, maltose, and sucrose) were determined using Association of Official Agricultural Chemists methods [15]. Specifically, the amount of protein was assessed using the micro-Kjeldahl method (N × 6.25). Ash was evaluated by calcining at 550 ± 10 °C for 5 h. Fats were measured using a Soxhlet apparatus. Carbohydrates were estimated using difference analysis. Glucose, fructose, lactose, maltose, and sucrose contents were determined using ion chromatography, and the energy value was calculated as follows:Energy (kcal/100 g dry weight [dw]) = 4 × (g/100 g dw proteins + g/100 g dw carbohydrates) + 9 × (g/100 g dw fats)(1)

### 2.4. Amino Acids

Proteins extracted from SOB or DSOB were hydrolyzed with hydrochloric acid at 110 °C for 21 h, and then filtered using a 0.22 μm membrane for injection into a high-performance liquid chromatography (HPLC) (UltiMate 3000; Thermo Fisher Scientific, Waltham, MA, USA) instrument equipped with a fluorescence detector (UltiMate 3000; Thermo Fisher Scientific). Separation was achieved using a ZORBAX Eclipse-AAA column (150 mm length, 3.0 mm i.d., and 3.5 μm particle size; Agilent, Santa Clara, CA, USA) at 35 °C. The samples (10 μL) were eluted with sodium dihydrogen phosphate solution (40 mM in water, pH 7.8, (A) and acetonitrile–methanol–water solution (45:45:10, *v*/*v*/*v*, (B) at 0.7 mL/min: 0% B at 0–1.9 min, 0–57% B at 1.9–18.1 min, 57–100% B at 18.1–18.6 min, and 100% B at 18.6–22.3 min. The excitation and emission wavelengths of the fluorescence detector were 340 nm and 450 nm, respectively. Amino acids were identified by comparing the retention times with those of the standards and were further quantified using calibration curves. Moreover, the chemical score (CS), essential amino acid index (EAAI), and biological value (BV) were calculated using the following equations:CS = EAA in sample protein/EAA in egg protein × 100(2)
EAAI = [(Lysines × Leucines × ··· × Histidines)/(Lysiner × Leuciner × ··· × Histidiner)]^1/n^(3)
where “EAA” is essential amino acids, “s” is essential amino acid in sample, “r” is essential amino acid in whole egg, and “n” is amount of amino acids (assuming “Phenylalanine + Tyrosine,” and “Methionine + Cysteine” are together).

### 2.5. Minerals

All samples were digested using a MARS6 microwave digester (CEM Corporation, Charlotte, NC, USA). The digested solutions were then filtered using 0.22 μm membranes. The filtrate was subsequently analyzed using an inductively coupled plasma-atomic emission spectrometer (Prodigy7; Teledyne Leeman Labs, Mason, OH, USA). The response signals were tracked at 766.49 nm (K), 589.59 nm (Na), 324.75 nm (Cu), 317.93 nm (Ca), 279.55 nm (Mg), 259.94 nm (Fe), 257.61 nm (Mn), and 206.20 nm (Zn). Each element was quantitatively analyzed in accordance with its standard calibrated curves. 

### 2.6. Tocopherols

Tocopherols in SOB and DSOB were determined using the UltiMate 3000 HPLC system with a UV detector (UltiMate 3000), as previously described with some modifications [16]. Briefly, 2 g of the samples were mixed with methanol (4 mL), *n*-hexane (4 mL), and saturated sodium chloride water solution (2 mL). The clear upper layer was collected and filtered for further HPLC analysis. The Eclipse Plus C18 column (150 × 4.6 mm, 3.5 μm; Agilent) was used for separation. Samples were eluted with methanol at 0.7 mL/min and detected at 300 nm. Tocopherols (α-, β-, and γ-isoforms) were confirmed and quantified using standard compounds with calibrated curves.

### 2.7. Organic Acids

Organic acids (oxalic acid and citric acid) in SOB and DSOB were extracted by meta-phosphoric acid and subsequently analyzed by HPLC-UV (UltiMate 3000) at 215 nm and 210 nm, respectively. Sodium dihydrogen phosphate solution (0.01 mol/L, pH 3.5) was used as the mobile phase. The eluent was monitored using an Ultimate AQ-C18 column (250 × 4.6 mm, 5 μm; Welch Materials, Shanghai, China) that was used for separation at 35 °C. Compounds were identified by comparison with standards and further quantified using calibration curves [16].

### 2.8. Total Phenolic Content (TPC) and Total Flavonoid Content (TFC)

SOB and DSOB were extracted twice by sonication at 30 °C with an ethanol aqueous solution (70%, *v*/*v*) for 30 min. The obtained extracts were concentrated at 37 °C using a rotary evaporator (RE-2000A; YARONG, Shanghai, China) under vacuum, lyophilized (FreeZone 4.5; Labconco, Kansas City, MO, USA), and stored at −20 °C for the determination of TPC, TFC, polyphenols, and antioxidant capacities. TPC was determined using the Folin–Ciocalteu colorimetric method and expressed as gallic acid equivalents (GAE) [17]. TFC was evaluated using the method reported by Kainama et al. [18] and expressed as (+)-catechin equivalents (CE).

### 2.9. Characterization of Polyphenols by Liquid Chromatography Coupled with Electrospray Ionization–Quadrupole Time of Flight–Mass Spectrometry Analysis (LC–ESI–QTOF/MS) 

The DSOB and SOB polyphenols were characterized using an LC–ESI–QTOF/MS instrument (XEVO G2, Waters, Milford, MA, USA) for tentative identification. Separation was accomplished at 30 °C using a CORTECS C18 column (50 × 2.1 mm, 1.6 μm; Waters). SOB or DSOB (4 μL) were eluted with acidified water (0.1% formic acid, (A) and organic solutions (acetonitrile/methanol = 3:1, *v*/*v*, (B) at a flow rate of 0.3 mL/min. The following were used as the optimized gradient chromatography conditions: 5–20% B at 0–2 min, 20–30% B at 2–5 min, 30–40% B at 5–7 min, 40–100% B at 7–10 min. Peak identification was performed in both negative and positive modes, and mass spectra in the *m*/*z* range of 50–1000 were obtained. The mass spectrometry conditions were as follows: source temperature, 120 °C; desolvation temperature, 450 °C; gas flow rate, 800 L/h; and nebulizer gas pressure, 6.5 bar. The capillary voltage was set to 2.0 kV. Data acquisition and analysis were performed using MassLynx V4.1 SCN 901 (Waters). The polyphenols were temporarily identified by comparing the literature and searching the library using UNIFI (Waters). The content ratio of polyphenols in DSOB and SOB were calculated by peak areas in TIC.

### 2.10. Antioxidant

The antioxidant capacities of SOB and DSOB were assessed using electron transfer assays (DPPH and ABTS) and ROS scavenging assays (HO•, NO, and ONOO^−^). The DPPH, ABTS, HO•, and NO scavenging assays were performed as reported by Kwon et al. [19]. The ONOO^−^ scavenging assay was carried out as reported by Hazra et al. [20]. BHT and Trolox were used as positive controls for the DPPH and ABTS assays, respectively. Ascorbic acid was used as the positive control for HO•, NO, and ONOO^−^ scavenging assays. The antioxidant activities were expressed as the percentage of radical elimination (%) and the sample concentration for 50% inhibition (IC_50_).

### 2.11. α-Glucosidase Inhibition Assay

The α-glucosidase inhibition assay was performed as previously reported [6]. Specifically, a mixture of phosphate-buffered saline (PBS, pH 7.4, 90 μL), sample solution (20 μL), *p*-nitrophenyl-α-D-glucopyranoside solution (30 μL, 1 mM in PBS), and α-glucosidase solution (60 μL, 15 μg/mL in PBS) were incubated at 37 °C for 30 min. Acabose was used as a positive control. Then, the absorbance of the incubated mixture was measured at 405 nm using a microplate reader (Synergy HTX, BioTek Instruments, Winooski, VT, USA). The inhibitory activity was expressed as the percentage inhibition (%) and the IC_50_.

### 2.12. Rat Lens Aldose Reductase Inhibition Assay

The rat lens aldose reductase inhibition assay of SOB and DSOB was performed as described previously [21]. In brief, the eye lens of Wistar rats (10 weeks old, weight 250–280 g) were collected and homogenized for further centrifugation. The rat lens aldose reductase remained in the supernatant. Then, 900 μL of a total mixture of NADPH (0.16 mM), ammonium sulfate (2.5 mM), DL-glyceraldehyde (2.5 mM), rat lens aldose reductase, and samples were incubated, and the activity of aldose reductase was determined by measuring the decrease in NADPH absorbance at 340 nm for 3 min using a spectrophotometer. Epalrestat was used as a positive control. The inhibitory activity was expressed as the percentage inhibition (%) and the IC_50_. 

### 2.13. Animal Care

All animal experimental procedures were conducted in accordance with the guidelines and approval of the Institutional Animal Care and Use Committee (IACUC) of Hebei University (IACUC-20180051). Prior to the experiment, rats were placed in a standardized laboratory environment under a 12 h light/12 h dark cycle at temperature of 20–26 °C and humidity of 40–70%. All animals had access to water and food. 

### 2.14. Statistical Analysis

All experiments were repeated at least in triplicate. Results are expressed as the mean ± standard deviation. Data were analyzed using SPSS Statistics software version 19.0 (IBM, Armonk, NY, USA). The comparison of mean values was performed using Student’s unpaired *t*-test or one-way analysis of variance, as required. *p* < 0.05 was considered significant.

## 3. Results and Discussion

### 3.1. α-Glucosidase Inhibition, Aldose Reductase Inhibition, and Antioxidant Capacity of DSOB

α-Glucosidase inhibitors from natural sources are often used as functional foods for intervening in postprandial hyperglycemia in patients with diabetes. Thus, the effect of DSOB on α-glucosidase inhibition was first investigated to assess its potential as a functional food ingredient in diabetes. As shown in Figure 1A, DSOB inhibited α-glucosidase activity by 1.47%, 72.41%, and 88.52% at concentrations of 1, 5, and 10 μg/mL, respectively. SOB decreased α-glucosidase activity by 2.52%, 69.33%, and 96.22% at 1, 5, and 10 μg/mL, respectively. The α-glucosidase inhibitory capacity of DSOB was lower than that of SOB at 10 μg/mL and showed no significant differences at 1 and 5 μg/mL. As shown in Table 1, DSOB had an IC_50_ of 3.31 μg/mL for α-glucosidase inhibition, which showed no significant difference compared to the IC_50_ of SOB (3.18 μg/mL). Interestingly, both DSOB and SOB showed dramatically higher α-glucosidase inhibitory capacities than acarbose. Grape seed and green tea are popular functional food ingredients that possess excellent α-glucosidase inhibitory activities. In a previous study, Yilmazer-Musa et al. reported that grape seed has an IC_50_ of 1.2 μg/mL for α-glucosidase inhibition, green tea has an IC_50_ of 0.5 μg/mL for α-glucosidase inhibition, and acarbose has an IC_50_ of 91 μg/mL for α-glucosidase inhibition [22]. Another study reported that fruiting body of *Phellinus merrillii* has an IC_50_ of 13.73 μg/mL for α-glucosidase inhibition [23]. These results implied that DSOB was a potent α-glucosidase inhibitor, in which the α-glucosidase inhibitory activity of DSOB was not affected by the defatting process from SOB. This work is the first report to indicate the α-glucosidase inhibitory activities of DSOB and SOB.

Recently, the development of naturally derived aldose reductase inhibitors with less toxicity as functional foods has attracted much attention. Our previous research has reported that SOB is a potent aldose reductase inhibitor [24], but the aldose reductase inhibitory activity of DSOB is unclear. Therefore, in the present study, the inhibitory effect of DSOB on aldose reductase was also investigated using rat lens to assess its potential as a functional food ingredient in diabetes. As shown in Figure 1B, DSOB decreases the catalyzing activities of aldose reductase by 23.79%, 72.33%, and 82.63% at concentrations of 1, 5, and 10 μg/mL, respectively. SOB decreased rat lens aldose reductase activities by 17.56%, 57.89%, and 79.70% at 1, 5, and 10 μg/mL, respectively. The inhibitory effects of DSOB at 1 and 5 μg/mL on rat lens aldose reductase were significantly higher than those of SOB. Moreover, DSOB has a significantly lower IC_50_ (2.56 μg/mL) than that of SOB (3.44 μg/mL), as shown in Table 1. Huang et al. reported that fruiting body of *Phellinus merrillii* has an IC_50_ of 12.55 μg/mL for aldose reductase inhibition [23]. Although the aldose reductase inhibitory activity of DSOB was much lower than that of epalrestat, these results indicated that DSOB was an excellent natural origin aldose reductase inhibitor, and the defatting process from SOB seemed to improve its inhibition. 

The antioxidant capacities of DSOB were further assessed using five assays, namely, O•, NO, ONOO^−^, DPPH, and ABTS, on the basis of the antioxidant mechanisms of electron transfer and ROS scavenging [19]. HO• is the main ROS in the body, inducing oxidative stress that breaks down DNA strands and damages proteins [25]. As shown in Table 2 and Figure 2, DSOB scavenges 45.18% HO• at 3.33 mg/mL. Although the HO• scavenging capacity of DSOB was significantly lower than that of SOB and ascorbic acid, these results implied that DSOB contained potent HO• scavengers. NO provides vascular protection; nevertheless, the highly reactive radical, ONOO^−^, is formed when NO interacts with superoxide radicals, leading to a series of harmful events [26]. DSOB showed good antioxidant activities against NO (IC_50_ = 1.08 μg/mL) and ONOO^−^ (IC_50_ = 512.49 μg/mL), with no significant difference with SOB. DSOB also showed excellent scavenging capacities for ABTS (IC_50_ = 1.22 μg/mL) and DPPH (IC_50_ = 15.62 μg/mL) radicals, with no significant difference compared to SOB. These findings demonstrated that DSOB is a good natural antioxidant.

### 3.2. Polyphenol Composition of DSOB

In recent years, an increase in the incidence of diabetes associated with lifestyle has led to a search for effective and safe methods of prophylaxis and/or therapy. The results of epidemiological studies indicate that diabetes may be prevented by enriching diets with plant foods, since these foods are rich in polyphenols with pharmacological activity [27,28]. Over the last few years, SOB has attracted wide attention because of its high polyphenol content. However, understanding of the polyphenols in DSOB remains limited. Therefore, the DSOB TPC and TFC were tested using SOB as a reference, and the results are listed in Table 3. The TPC of DSOB was 48.91 mg GAE/g dw, significantly higher than that of SOB (33.10 mg GAE/g dw). The TPC of grape seed and green tea were 86 and 74 mg GAE/g dw, respectively [22]. The amount of TFC in DSOB (33.93 mg CE/g dw) was also significantly higher than that in SOB (21.97 mg CE/g dw). These findings indicated that DSOB was a bulky reservoir of polyphenols. Therefore, the aforementioned remarkable bioactivities of SOB may be associated with its high phenolic and flavonoid content; the polyphenol composition of DSOB needs to be further illustrated. 

To better understand the polyphenol composition of DSOB, we performed LC–ESI–QTOF/MS because it provides high-resolution mass data containing affluent structural information of the analyte. In this work, 25 components in DSOB have been identified, including (+)-catechin, ellagic acid, gallic acid, procyanidins, quercetin, syringic acid, and their derivatives, as listed in Table 4. Among the 25 compounds, 14 compounds were reported to be α-glucosidase inhibitors, 13 compounds were reported to be aldose reductase inhibitors, and 21 compounds were reported to be antioxidants. Interestingly, although the profiles of the polyphenol composition in DSOB and SOB were similar (Figure 3), the quantities of each component in DSOB and SOB were quite different (Table 4). Among the 25 compounds, the amounts of syringic anhydride, trimethylenglykol-digalloat, monogalloylglucose, ellagic acid glycoside, ellagic acid xyloside, syringic acid, ellagic acid, quercetin glucuronide, and kaempferol glucuronide were higher in DSOB than those in SOB. The amounts of digalloylglucose, galloylxy trihydroxyflavanone, (+)-catechin, catechin gallate, methyl ellagic acid, procyanidins, and procyanidin B gallate were lower in DSOB than those in SOB. The differences in biological activities between DSOB and SOB may be caused by differences in the content of these polyphenols.

### 3.3. Nutritional Value of DSOB 

The nutritional and energy values of DSOB are presented as dw in Table 5, using SOB as a reference. Carbohydrates (72.60 g/100 g dw) were the main nutrient in DSOB, followed by protein (15.86 g/100 g dw), ash (8.86 g/100 g dw), and fats (2.68 g/100 g dw). However, carbohydrate, protein, ash, and fat content in SOB were 56.34, 12.80, 7.00, and 23.86 g/100 g dw, respectively. All nutritional values, except fats, were higher in DSOB than those in SOB. Seeds are normally a fat reservoir; thus, fats are the major macronutrients in SOB, ranking only second to carbohydrates. Compared to SOB, the proportion of fats in DSOB was greatly reduced by oil extraction, which led to decreased biomass and thus an increased nutritional content (carbohydrates, proteins, and ash) in disguise. The energy of SOB was 491.33 Kcal/100 g dw, calculated on the basis of its carbohydrate, protein, and fat content. A lower energy was observed for DSOB (378.00 Kcal/100 g dw) due to its lower fat content, since fats are the main contributor to energy. 

The amount and type of carbohydrates to eat is an important consideration for patients with diabetes when planning their diet. As shown in Table 6, the predominant sugars in DSOB were found to be sucrose (1.27 g/100 g dw) and glucose (0.16 g/100 g dw), whereas other sugars were not detected. Briefly, there was 0.13 g/100 g dw maltose in SOB, and the contents of sucrose (0.99 g/100 g dw) and glucose (0.13 g/100 g dw) in SOB were significantly lower than those in DSOB. However, both DSOB and SOB presented low levels of monosaccharides and disaccharides. Furthermore, a higher fiber content was observed in DSOB (34.78 g/100 g dw) compared to that in SOB (29.40 g/100 g dw). These results demonstrated that DSOB had a lower glycemic index, thereby reducing the health risks induced by sugars.

As mentioned previously, SOB is a good source of protein and, interestingly, DSOB has a higher protein content. As shown in Table 7, non-essential amino acids in DSOB proteins are completely dominant. In particular, glutamic acid (14.24 g/100 g protein) was highest in DSOB, followed by arginine (6.72 g/100 g protein), aspartic acid (6.16 g/100 g protein), proline (5.76 g/100 g protein), glycine (4.88 g/100 g protein), serine (4.32 g/100 g protein), and alanine (2.72 g/100 g protein). Regarding essential amino acids, leucine (4.40 g/100 g protein) had the highest levels, followed by phenylalanine (3.52 g/100 g protein), valine (3.12 g/100 g protein), threonine (2.32 g/100 g protein), lysine (2.32 g/100 g protein), isoleucine (1.68 g/100 g protein), cysteine (1.52 g/100 g protein), tyrosine (1.52 g/100 g protein), histidine (1.28 g/100 g protein), and methionine (1.09 g/100 g protein). Overall, the amino acid content of DSOB was less than that of SOB but exhibited no significant differences. DSOB protein quality was further assessed by CS, EAAI, and BV. The CS of the essential amino acids in DSOB, based on egg protein, showed values between 31.11 and 58.18, and Iso was the first restricted amino acid; the EAAI of 44.84 and BV of 37.15 in DSOB indicated that overall levels of essential amino acids were relatively low.

As shown in Table 8, the amounts of four macroelements (Na, K, Ca, and Mg) and four microelements (Fe, Mn, Cu, and Zn) in DSOB and SOB were determined. Among the macroelements, Na (1121.67 mg/100 g dw) was present in the highest amount in DSOB, followed by Ca (1106.77 mg/100 g dw), Mg (320.88 mg/100 g dw), and K (259.63 mg/100 g dw). Regarding the microelements, Fe (50.35 mg/100 g dw) exhibited the highest level, followed by Mn (7.22 mg/100 g dw), Zn (3.6 mg/100 g dw), and Cu (0.40 mg/100 g dw). Overall, the DSOB mineral content was significantly higher than that of SOB, except for K and Cu. The daily requirement of macroelements is approximately 100 mg to maintain activity. However, only a few milligrams, even micrograms, of microelements are required per day, as their name suggests. Thus, DSOB was a good source of minerals for dietary supplementation, especially Ca and Fe. 

Two organic acids (oxalic acid and citric acid) were identified in DSOB and SOB. As shown in Table 9, oxalic acid levels in DSOB are 255.30 mg/100 g dw, which is significantly lower than that in SOB (584.65 mg/100 g dw). In contrast, citric acid in DSOB (363.26 mg/100 g dw) was significantly greater than that in SOB (171.67 mg/100 g DW). Organic acids are a key indicator of food quality and serve as synergists to improve the activity of antioxidants. However, oxalic acid is an anti-nutrient whose high dietary intake results in the formation of calcium oxalate crystals, decreasing the absorption of Ca and increasing the risk of kidney stones. From this point of view, DSOB was more applicable for dietary supplementation than SOB.

The tocopherol contents (α-, β-, and γ-isoforms) have been determined, as shown in Table 9. Only one isoform of tocopherol (γ-) was identified and quantified in DSOB (1.11 mg/100 g dw) and SOB (11.61 mg/100 dw). Compared to SOB, the low tocopherol content in DSOB was likely related to its low fat content.

## 4. Conclusions

In summary, DSOB was found to be a reservoir of nutrients and polyphenols, and had remarkable inhibitory activity for α-glucosidase and aldose reductase, as well as antioxidant capacities. Moreover, 25 compounds in DSOB were temporarily identified using LC–ESI–QTOF/MS analysis, which were associated with the remarkable bioactivities of DSOB. DSOB also contained a high content of carbohydrates, proteins, and minerals. The fat content in DSOB decreased after EPO extraction from SOB, resulting in low levels of γ-tocopherol and energy. DSOB contained large amounts of fiber and low levels of sugars, providing a low glycemic index. Notably, DSOB was a good source of Ca and Fe, and the anti-nutrient and oxalic acid content was low. These findings imply that DSOB, that is, the waste material generated by SOB after EPO extraction, has potential as a functional food ingredient in diabetes. 

## Figures and Tables

**Figure 1 foods-10-00538-f001:**
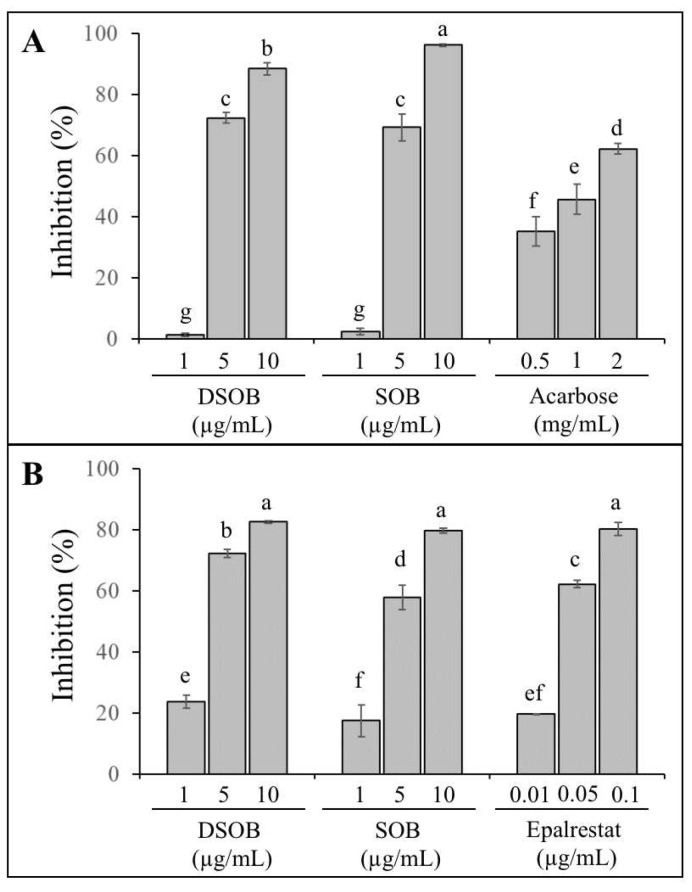
The inhibitory effects of defatted seeds of *Oenothera biennis* (DSOB) and seeds of *Oenothera biennis* (SOB) on α-glucosidase (**A**) and rat lens aldose reductase (**B**). Data in the bar graph with different letters are significantly different from each other (*p* < 0.05).

**Figure 2 foods-10-00538-f002:**
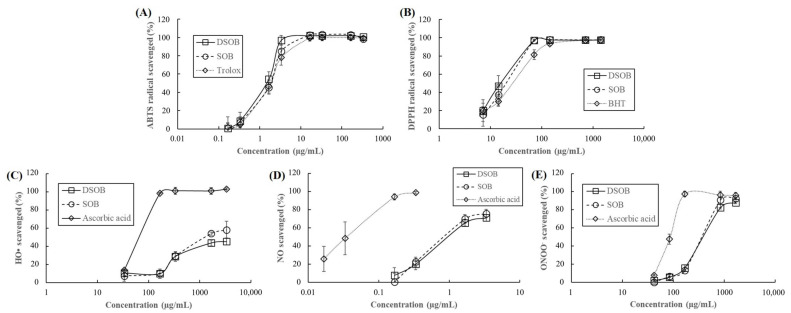
Antioxidant capacities of defatted seeds of *Oenothera biennis* (DSOB) and seeds of *Oenothera biennis* (SOB). (**A**) 2,2-Azinobis(3-ethylbenzothiazoline-6-sulfonic acid) diammonium salt (ABTS) radical scavenging activity of DSOB, SOB, and Trolox (positive control). (**B**) 1,1-Diphenyl-2-picrylhydrazyl (DPPH) radical scavenging activity of DSOB, SOB, and 2,6-di-tert-butyl-4-methylphenol (BHT, positive control). (**C**) Hydroxyl radical (HO•) scavenging activity of DSOB, SOB, and ascorbic acid (positive control). (**D**) Nitric oxide (NO) scavenging activity of DSOB, SOB, and ascorbic acid (positive control). (**E**) Peroxynitrite (ONOO^−^) scavenging of DSOB, SOB, and ascorbic acid (positive control).

**Figure 3 foods-10-00538-f003:**
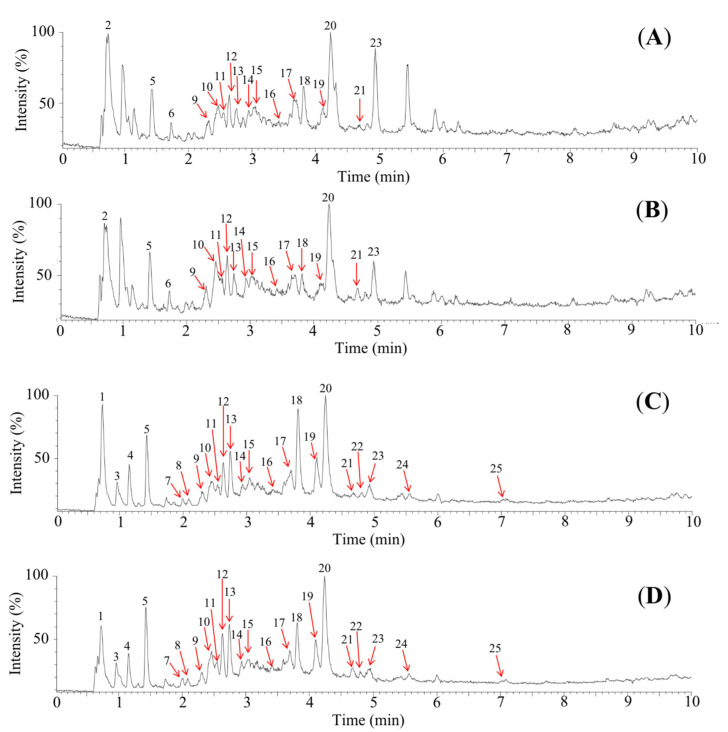
Total ion chromatography of 70% ethanol extracts from defatted seeds of *Oenothera biennis* ((**A**) in positive mode, (**C**) in negative mode) and seeds of *Oenothera biennis* ((**B**) in positive mode, (**D**) in negative mode).

**Table 1 foods-10-00538-t001:** The IC_50_ values of defatted seeds of *Oenothera biennis* (DSOB) and seeds of *Oenothera biennis* (SOB).

Sample	IC_50_ (μg/mL) ^1^	
	α-Glucosidase	Rat Lens Aldose Reductase
DSOB	3.31 ± 0.09 ^a 2^	2.56 ± 0.06 ^a^
SOB	3.18 ± 0.13 ^a^	3.44 ± 0.39 ^b^
Positive control ^3^	1265.7 ± 29.4 ^b^	0.031 ± 0.001 ^c^

^1^ IC_50_ is the sample concentration providing 50% inhibition. ^2^ Results are presented as the mean ± standard deviation (*n* = 3). Values within a column of the table marked with different letters are significantly different from each other (*p* < 0.05). ^3^ Acarbose was used as positive control in α-glucosidase assay; epalrestat was used as positive control in rat lens aldose reductase assay.

**Table 2 foods-10-00538-t002:** IC_50_ of defatted seeds of *Oenothera biennis* (DSOB) and seeds of *Oenothera biennis* (SOB) on radical scavenging.

Samples	IC_50_ ^1^ (μg/mL)				
	DPPH ^2^	ABTS ^3^	HO• ^4^	NO ^5^	ONOO^− 6^
DSOB	15.62 ± 4.79 ^a 7^	1.12 ± 0.11 ^a^	Na ^8^	1.08 ± 0.20 ^a^	512.49 ± 9.64 ^a^
SOB	18.04 ± 5.20 ^ab^	1.43 ± 0.30 ^a^	1784.01 ± 475.14 ^a^	1.01 ± 0.22 ^a^	470.54 ± 49.68 ^b^
Positive control	33.12 ± 11.93 ^b^	1.34 ± 0.07 ^a^	110.06 ± 11.38 ^b^	0.039 ± 0.019 ^b^	94.99 ± 0.23 ^c^

^1^ IC_50_ is the sample concentration providing 50% radical scavenging. ^2^ DPPH is 1,1-diphenyl-2-picrylhydrazyl. 2,6-di-tert-Butyl-4-methylphenol was used as positive control. ^3^ ABTS is 2,2-azinobis(3-ethylbenzothiazoline-6-sulfonic acid) diammonium salt. Trolox was used as positive control. ^4^ HO• is hydroxyl radical. Ascorbic acid was used as positive control. ^5^ NO is nitric oxide radical. Ascorbic acid was used as positive control. ^6^ ONOO^−^ is peroxynitrite. Ascorbic acid was used as positive control. ^7^ Results are presented as the mean ± standard deviation (*n* = 3). Values within a column superscripted with different letters are significantly different from each other (*p* < 0.05). ^8^ “Na” indicates not active.

**Table 3 foods-10-00538-t003:** Total phenolic content (TPC) and total flavonoid content (TFC) of defatted seeds of *Oenothera biennis* (DSOB) and seeds of *Oenothera biennis* (SOB).

	DSOB	SOB	*p*-Value ^4^
TPC (mg GAE/g dw) ^1^	48.91 ± 2.08 ^3^	33.10 ± 2.27	<0.001
TFC (mg CE/g dw) ^2^	33.93 ± 0.60	21.97 ± 0.56	<0.001

^1^ Results of TPC are expressed as milligrams of gallic acid equivalents (GAE) per gram dry weight of DSOB or SOB; “dw” is short for dry weight. ^2^ Results of TFC are expressed as milligrams of (+)-catechin equivalents (CE) per gram dry weight of DSOB or SOB. ^3^ Results are presented as the mean ± standard deviation (*n* = 3). ^4^
*p*-value was calculated by Student’s unpaired *t*-test. *p* < 0.05 is considered significant.

**Table 4 foods-10-00538-t004:** Polyphenol compounds detected and tentatively characterized in defatted seeds of *Oenothera biennis* (DSOB) and seeds of *Oenothera biennis* (SOB) using liquid chromatography coupled with electrospray ionization and quadrupole time of flight mass spectrometry (LC–ESI–QTOF/MS) in positive and negative ion modes.

	*t*_R_ (min) ^1^	[M+H]^+^ *m*/*z*	[M-H]^−^ *m*/*z*	Predicted Formula	Observed (*m*/*z*)	Theroetical (*m*/*z*)	Mass Error (mDa)	DBE ^2^	Temporarily Identified	Activity	Content Ratio (DSOB/SOB)
1	0.70	- ^3^	377.0824	C_18_H_18_O_9_	377.0824	377.0873	−4.9	10.5	Syringic anhydride	-	4.47
2	0.72	381.0837	-	C_17_H_16_O_10_	381.0837	381.0822	1.5	9.5	Trimethylenglykol digalloat	-	3.05
3	0.93	-	331.0713	C_13_H_16_O_10_	331.0713	331.0665	4.8	6.5	Monogalloylglucose	α-Glucosidase inhibitor [29]; antioxidant [30]	1.06
4	1.13	-	331.0634	C_13_H_16_O_10_	331.0634	331.0665	−3.1	6.5	Monogalloylglucose	α-Glucosidase inhibitor [29]; antioxidant [30]	3.38
5	1.40	171.0356	169.0145	C_7_H_6_O_5_	169.0145	169.0137	0.8	5.5	Gallic acid	α-Glucosidase inhibitor [31]; aldose reductase inhibitor [24]; antioxidant [32]	1.04
6	1.71	449.0734	-	C_20_H_16_O_12_	449.0734	449.0720	1.4	12.5	Methyl ellagic acid xyloside	-	1.01
7	1.98	-	483.0811	C_20_H_20_O_14_	483.0811	483.0775	3.6	11.5	Digalloylglucose	α-Glucosidase inhibitor [33]; aldose reductase inhibitor [34]; antioxidant [35]	0.88
8	2.07	-	153.0180	C_7_H_6_O_4_	153.0180	153.0188	−0.8	5.5	Protocatechuic acid	α-Glucosidase inhibitor [36]; aldose reductase inhibitor [37]; antioxidant [38]	1.03
9	2.29	579.1470	577.1298	C_30_H_26_O_12_	579.1470	579.1503	−3.3	17.5	Procyanidin B	α-Glucosidase inhibitor [39]; aldose reductase inhibitor [24]; antioxidant [40]	0.87
10	2.44	579.1470	577.1403	C_30_H_26_O_12_	579.1470	579.1503	−3.3	17.5	Procyanidin B	α-Glucosidase inhibitor [39]; aldose reductase inhibitor [24]; antioxidant [40]	0.71
11	2.52	867.2014	865.1972	C_45_H_38_H_18_	865.1972	865.1980	−0.8	27.5	Procyanidin trimer	α-Glucosidase inhibitor [41]; antioxidant [42]	0.60
12	2.62	291.0899	289.0694	C_15_H_14_O_6_	289.0694	289.0712	−1.8	9.5	(+)-Catechin	α-Glucosidase inhibitor [43]; aldose reductase inhibitor [44]; antioxidant [45]	0.86
13	2.72	185.0494	183.0301	C_8_H_8_O_5_	183.0301	183.0293	0.8	5.5	Methyl gallate	α-Glucosidase inhibitor [46]; aldose reductase inhibitor [24]; antioxidant [47]	0.92
14	2.90	731.1622	729.1451	C_37_H_30_O_16_	729.1451	729.1456	−0.5	23.5	Procyanidin B gallate	Antioxidant [48]	0.82
15	3.04	465.0714	463.0554	C_20_H_16_O_13_	463.0554	463.0513	4.1	13.5	Ellagic acid glycoside	Aldose reductase inhibitor [49]; antioxidant [50]	1.67
16	3.39	443.0980	441.0837	C_22_H_18_O_10_	443.0980	443.0978	0.2	13.5	Catechin gallate	Aldose reductase inhibitor [51]; antioxidant [52]	0.79
17	3.63	435.0565	433.0388	C_19_H_14_O_12_	435.0565	435.0564	0.1	12.5	Ellagic acid xyloside	Antioxidant [53]	1.37
18	3.79	199.0663	197.0456	C_9_H_10_O_5_	197.0456	197.0450	0.6	5.5	Syringic acid	α-Glucosidase inhibitor [54]; aldose reductase inhibitor [55]; antioxidant [56]	1.87
19	4.09	303.0162	300.9980	C_14_H_6_O_8_	300.9980	300.9984	−0.4	12.5	Ellagic acid	α-Glucosidase inhibitor [57]; aldose reductase inhibitor [58]; antioxidant [59]	1.13
20	4.23	479.0820	477.0708	C_21_H_18_O_13_	479.0820	479.0826	−0.6	12.5	Quercetin glucuronide	α-Glucosidase inhibitor [60]; antioxidant [61]	1.18
21	4.67	441.0876	439.0679	C_22_H_16_O_10_	439.0679	439.0665	1.4	15.5	Galloylxy trihydroxyflavanone	-	0.05
22	4.79	-	433.0752	C_20_H_18_O_11_	433.0752	433.0771	−1.9	12.5	Quercetin xylopyranoside	Antioxidant [62]	1.15
23	4.92	463.0877	461.0743	C_21_H_18_O_12_	463.0877	463.0877	−0.8	12.5	Kaempferol glucuronide	Antioxidant [63]	2.37
24	5.54	-	315.0179	C_15_H_8_O_8_	315.0179	315.0141	3.8	12.5	Methyl ellagic acid	α-Glucosidase inhibitor [64]; aldose reductase inhibitor [2]; antioxidant [65]	0.88
25	6.99	303.0542	301.0359	C_15_H_10_O_7_	301.0359	301.0348	1.1	11.5	Quercetin	α-Glucosidase inhibitor [66]; aldose reductase inhibitor [67]; antioxidant [68]	1.08

^1^*t*_R_, retention time. ^2^ DBE, double bond equivalency. ^3^ “-” indicates not detected.

**Table 5 foods-10-00538-t005:** Nutritional value and energetic value of defatted seeds of *Oenothera biennis* (DSOB) and seeds of *Oenothera biennis* (SOB).

Samples	Carbohydrates (g/100 g dw ^1^)	Proteins (g/100 g dw)	Ash (g/100 g dw)	Fat (g/100 g dw)	Energy (Kcal/100 g dw)
DSOB	72.60 ± 1.12 ^2^	15.86 ± 0.95	8.86 ± 0.24	2.68 ± 0.89	378.00 ± 4.74
SOB	56.34 ± 1.96	12.80 ± 0.77	7.00 ± 0.03	23.86 ± 1.16	491.33 ± 5.68
*p*-Value ^3^	0.002	0.013	0.001	<0.001	<0.001

^1^ “dw” is short for dry weight. ^2^ Results are presented as the mean ± standard deviation (*n* = 3). ^3^
*p*-value was calculated by Student’s unpaired *t*-test. *p* < 0.05 was considered significant.

**Table 6 foods-10-00538-t006:** The contents of fibers and sugars in defatted seeds of *Oenothera biennis* (DSOB) and seeds of *Oenothera biennis* (SOB).

Sample	DSOB	SOB	*p*-Value ^1^
Glucose (g/100 g dw ^2^)	0.16 ± 0.010 ^3^	0.13 ± 0.006	0.034
Fructose (g/100 g dw)	ND ^4^	ND	- ^5^
Galactose (g/100 g dw)	ND	ND	-
Sucrose (g/100 g dw)	1.27 ± 0.08	0.99 ± 0.04	0.042
Maltose (g/100 g dw)	ND	0.13 ± 0.005	<0.001
Lactose (g/100 g dw)	ND	ND	-
Fibers (g/100 g dw)	34.78 ± 2.50	29.40 ± 1.28	0.03

^1^*p*-value was calculated by Student’s unpaired *t*-test. *p* < 0.05 was considered significant. ^2^ “dw” is short for dry weight. ^3^ Results are presented as the mean ± standard deviation (*n* = 3). ^4^ “ND” indicates not detected. ^5^ “-” indicates not available.

**Table 7 foods-10-00538-t007:** Amino acid composition (%) of proteins in defatted seeds of *Oenothera biennis* (DSOB) and seeds of *Oenothera biennis* (SOB).

Amino Acids		Amino Acid (g/100 g Protein)		*p*-Value ^2^	CS ^3^		*p*-Value
		DSOB	SOB		DSOB	SOB	
Essential amino acid							
	Valine	3.12 ± 0.19 ^1^	3.28 ± 0.20	0.493	47.27 ± 2.84	49.70 ± 2.98	0.487
	Isoleucine	1.68 ± 0.10	1.72 ± 0.10	0.730	31.11 ± 1.87	31.85 ± 1.91	0.735
	Leucine	4.40 ± 0.26	4.69 ± 0.28	0.384	51.16 ± 3.07	54.53 ± 3.27	0.388
	Threonine	2.32 ± 0.14	2.50 ± 0.15	0.324	49.36 ± 2.96	53.19 ± 3.19	0.323
	Lysine	2.32 ± 0.14	2.60 ± 0.16	0.175	33.14 ± 1.99	37.14 ± 2.23	0.170
	Histidine	1.28 ± 0.08	1.48 ± 0.09	0.112	58.18 ± 3.49	67.27 ± 4.04	0.106
	Tyrosine	1.52 ± 0.09	1.49 ± 0.09	0.773	54.19 ± 3.25	55.48 ± 3.33	0.736
	Phenylalanine	3.52 ± 0.21	3.67 ± 0.22	0.555			
	Cysteine	1.52 ± 0.09	1.47 ± 0.09	0.635	42.11 ± 2.53	44.91 ± 2.69	0.383
	Methionine	0.88 ± 0.05	1.09 ± 0.07	0.047			
Non-essential amino acid							
	Arginine	6.72 ± 0.40	7.27 ± 0.44	0.303			
	Proline	5.76 ± 0.35	6.71 ± 0.40	0.096			
	Aspartic	6.16 ± 0.37	6.72 ± 0.40	0.262			
	Serine	4.32 ± 0.26	4.61 ± 0.28	0.384			
	Glutamic acid	14.24 ± 0.85	15.63 ± 0.94	0.238			
	Glycine	4.88 ± 0.29	5.23 ± 0.31	0.349			
	Alanine	2.72 ± 0.16	2.81 ± 0.17	0.641			
EAAI ^4^		44.84 ± 2.69	48.09 ± 2.89	0.350			
BV ^5^		37.15 ± 2.93	40.68 ± 3.14	0.350			

^1^ Results are presented as the mean ± standard deviation (*n* = 3). ^2^
*p*-value was calculated by Student’s unpaired *t*-test. *p* < 0.05 was considered significant. ^3^ “CS” indicates chemical score. ^4^ “EAAI” indicates essential amino acid index. ^5^ “BV” indicates biological value.

**Table 8 foods-10-00538-t008:** The contents of minerals in defatted seeds of *Oenothera biennis* (DSOB) and seeds of *Oenothera biennis* (SOB).

Sample	DSOB	SOB	*p*-Value ^3^
Macroelements (mg/100 g dw ^1^)			
Na	1121.67 ± 3.45 ^2^	999.44 ± 6.20	<0.001
K	259.63 ± 1.50	303.67 ± 0.87	<0.001
Ca	1106.77 ± 7.73	1001.69 ± 7.08	<0.001
Mg	320.88 ± 1.34	243.74 ± 1.01	<0.001
Microelements (mg/100 g dw)			
Fe	50.35 ± 0.63	42.86 ± 0.25	<0.001
Mn	7.22 ± 0.04	5.72 ± 0.12	<0.001
Cu	0.40 ± 0.003	0.43 ± 0.008	0.016
Zn	3.60 ± 0.001	2.02 ± 0.05	<0.001

^1^ “dw” is short for dry weight. ^2^ Results are presented as the mean ± standard deviation (*n* = 3). ^3^
*p*-value was calculated by Student’s unpaired *t*-test. *p* < 0.05 was considered significant.

**Table 9 foods-10-00538-t009:** The contents of oxalic acid, citric acid, and tocopherols in defatted seeds of *Oenothera biennis* (DSOB) and seeds of *Oenothera biennis* (SOB).

Sample	DSOB	SOB	*p*-Value ^3^
Oxalic acid (mg/100 g dw ^1^)	255.30 ± 30.00 ^2^	584.65 ± 52.23	0.002
Citric acid (mg/100 g dw)	363.26 ± 18.05	171.67 ± 22.67	0.001
α-Tocopherol (mg/100 g dw)	ND ^4^	ND	- ^5^
β-Tocopherol (mg/100 g dw)	ND	ND	-
γ-Tocopherol (mg/100 g dw)	1.11 ± 0.001	11.61 ± 0.04	<0.001

^1^ “dw” is short for dry weight. ^2^ Results are presented as the mean ± standard deviation (*n* = 3). ^3^
*p*-value was calculated by Student’s unpaired *t*-test. *p* < 0.05 was considered significant. ^4^ “ND” indicates not detected. ^5^ “-“ indicates not available.

## Data Availability

The study did not report any data.

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
