# Peer review of "Defatted Seeds of *Oenothera biennis* as a Potential Functional Food Ingredient for Diabetes"

_foods, 2021, doi:10.3390/foods10030538_

Round 1
Reviewer 1 Report
The authors have done a very good job. The research is of interest to the readership of the journal and the methods used are explained in full. The authors have tried to critically evaluate their results. I would suggest that they need to look at incorporating some newer references (2020 and 2021 references) so as to illustrate the relevance of their work to current literature.
Well done, I enjoyed reading the manuscript.
Reviewer 2 Report
A great publications on the valorization of co-products in the field of health nutritional applications. In this case about seeds of evening primrose oil after defatting seeds. These initial informations are complete and clear. However I just regret a lack of informations about the stability of the products.
Reviewer 3 Report
The paper is generally interesting.
However, a major revision is needed.
In details:
- no discussion of the obtained results is present. The Authors should compare the detected bioactivities and composition of DSOB with other functional food with similar properties
- -How did the Authors identify the chemical structure of polyphenols? By comapring experimental data with literature data?
- how did the Authors perform the quantificaton of polyphenols? Better explain
- Figire 1 contain a table which should be separated
Round 2
Reviewer 3 Report
Now the manuscript is suitable for publication